# Anisotropic Thermal Conductivity in Few-Layer and Bulk Titanium Trisulphide from First Principles

**DOI:** 10.3390/nano10040704

**Published:** 2020-04-08

**Authors:** Fernan Saiz, Jesus Carrete, Riccardo Rurali

**Affiliations:** 1Institut de Ciència de Materials de Barcelona (ICMAB–CSIC) Campus de Bellaterra, Bellaterra, 08193 Barcelona, Spain; fsaiz@icmab.es; 2Institute of Materials Chemistry, TU Wien, A-1060 Vienna, Austria; jesus.carrete.montana@tuwien.ac.at

**Keywords:** thermal transport, first-principles calculations, lattice dynamics, 2D materials

## Abstract

We study the thermal conductivity of monolayer, bilayer, and bulk titanium trisulphide (TiS${}_{3}$) by means of an iterative solution of the Boltzmann transport equation based on *ab-initio* force constants. Our results show that the thermal conductivity of these layers is anisotropic and highlight the importance of enforcing the fundamental symmetries in order to accurately describe the quadratic dispersion of the flexural phonon branch near the center of the Brillouin zone.

## 1. Introduction

The study of two-dimensional materials is a very active research field, as these compounds exhibit enhanced properties with respect to those shown by their bulk counterparts. For example, the thermal conductivity of graphene [1,2] has been reported to be as high as 5300 W m${}^{-1}$K${}^{-1}$, which is an order of magnitude higher than graphite’s [3]. Beyond graphene, the properties of new two-dimensional materials are being investigated as new methods to exfoliate flakes with thicknesses of several nanometers are being developed. Among these materials, transition metal trichalcogenides (MX${}_{3}$ with M = Ti, Zr, or Hf and X = S, Se, or Te) have attracted much attention due to their narrow band gap of around 1.0 eV [4,5], which makes them not only suitable for optical and electronic properties [6], but also for thermoelectric applications [7]. Furthermore, the structure of these chalcogenides is anisotropic, leading to different transport coefficients in the flake plane, from which undesirable backscattering from hot electrons can be minimized [8].

Within this trichalcogenide family, recent studies have shown that titanium trisulphide (TiS${}_{3}$) is an excellent candidate to build electronic and optic nanodevices [6], electrodes for Li and Na ion batteries [9], anodes for hydrogen photogeneration [10], and photodetectors [11,12,13] due to a direct band gap of 1.0 ± 0.1 eV for thin films with a thickness of around 300 nm [14]. TiS${}_{3}$ is an *n*-type semiconducting material formed by abundant and low toxicity elements that is also attractive for thermoelectric applications because of its large Seebeck coefficient *S* of −650 μV/K at 300 K and a low thermal conductivity, κ, of 3.7 W m${}^{-1}$K${}^{-1}$ at 325 K in bulk form [15]. Similar measurements have reported a thermopower of nearly −700 μV/K [16] at 300 K with an electrical conductivity σ at around 200 K between 0.5 and 5.0 (Ω cm)${}^{-1}$. The efficiency of a thermoelectric material is characterized by the figure of merit, which is calculated for a given temperature *T* as
(1)$$zT=\frac{S^{2}\sigma }{\kappa }T,$$
where σ and κ are the electrical and thermal conductivities, and *S* is the Seebeck coefficient. Therefore, any degree of anisotropy in all transport properties present in Equation (Equation 1) can be exploited to optimize the conversion of heat into electricity in these compounds. In particular, we ask ourselves if anisotropy can be a degree of freedom to increase the efficiency of few-layer and bulk TiS${}_{3}$, particularly where the reduction of the thermal conductivity is concerned. Estimates of the thermal properties of monolayer TiS${}_{3}$ from first-principles have been previously reported by Zhang et al. [17]. In this work, however, we also compute the thermal conductivity of bilayers and of the bulk form of this material to help fill the gap in the experimental data between the well-established results for the single layer and for (bulk) multilayers.

The goal of this work is then to calculate the thermal conductivity tensor of monolayers and bilayers of TiS${}_{3}$, as well as that of its bulk phase. This calculation is performed employing a first-principles approach whereby density functional theory (DFT) is used to relax the geometry of these TiS${}_{3}$ structures to then predict their thermal properties with a lattice dynamics scheme. Our intention here is to track the evolution of the thermal conductivity tensor as the thickness of a nanosheet is doubled and compare it with that obtained in the bulk, as limit of a thick multilayer. In this work we pay special attention to the correct enforcement of the translation and rotation symmetries of free space for the two nanosheets and to their effect on the thermal conductivity. Conserving these symmetries requires correcting the second-order force constants by defining a set of internal coordinates for the monolayer and bilayer that span the same space as the original Cartesian coordinates of the crystalline system except for rigid translations and rotations. This has been shown to lead to a correct description of out-of-plane vibrations [18]. Therefore, we apply the correction formulated in Ref. [18], where it was found that by enforcing these symmetries in the case of borophene, its computed thermal conductivity decreases by 50% and reverses its anisotropy as well as that the lowest-lying phonon branch becomes quadratic in the neighborhood of the Γ point in reciprocal space.

The manuscript is organized as follows: in Section 2 we describe the recipe used to run our first-principles calculations, starting from the geometry optimisations of the TiS${}_{3}$ systems to then compute the thermal properties with lattice dynamics; next, in Section 3, we compare the phonon dispersions of these systems before and after correcting the interatomic force constants and we analyse the impact of this correction on the anisotropy and values of the thermal conductivity tensor; finally, Section 4 concludes with our main findings and the implications of this work.

## 2. Theoretical Methods

Our methodology starts by employing the Vienna Ab Initio Simulation Package (VASP) [19,20,21,22] to relax the positions and cell parameters of these nanosheets. The monolayers are first represented as a unit cell belonging to the P2${}_{1/m}$ space group as shown in Figure 1. Reciprocal space integrations are performed using a mesh of 10×14×1
**k**-points centered at Γ in the Brillouin zone. The geometry relaxation is carried out setting thresholds of 0.01 eV/nm for the forces and $1\times 10^{-6}$ eV for the self-consistent solution of the wavefunction. We use the generalized gradient approximation to the exchange-correlation potential in the Perdew-Burke-Ernzerhof (PBE) flavour [23] and the projector augmented wave method (PAW) [24,25]. We expand the valence orbitals on a plane wave basis with an energy cutoff of 350 eV. Once the geometry of the TiS${}_{3}$ monolayer is optimized, we obtain the following parameters for the monolayer: a=0.496 nm, b=0.339 nm; for the bilayer: a=0.495 nm, b=0.340 nm; for the bulk: a=0.494 nm, b=0.339 nm, c=0.887, and β= 96.10 degrees. As a reference, bulk values from Ref. [26] are a=0.458 nm, b=0.340 nm, c=0.878 nm, and β= 97.32 degrees. Long-range van der Waals forces are included using the zero damping Grimme DFT-D3 scheme [27]. We choose this method because it yields an interlayer separation of 0.313 nm, in better agreement with the bulk value than the predictions of other van der Waals forces descriptions available in VASP.

Starting from the optimized primitive cell, we compute the interatomic force constants (IFCs) in 4×5 supercells for the monolayer and the bilayer and 4×5×3 supercells for the bulk using a finite differences method. For the harmonic displacements we use the Phonopy code [28] considering all neighboring interactions. thirdorder.py [29,30] is used to characterize the anharmonic interactions, neglecting those beyond seventh neighbors. The IFCs are then used as an input to solve the Boltzmann Transport Equation (BTE) iteratively with the almaBTE code [31] and the lattice thermal conductivity is obtained as
(2)$$\kappa_{ij}=\sum_{\lambda }\kappa_{ij,\lambda }=C\sum_{\lambda }f_{\lambda }\left(f_{\lambda }+1\right){\left(h\nu_{\lambda }\right)}^{2}v_{i,\lambda }F_{j,\lambda },$$
where *i* and *j* are the spatial directions *x*, *y*, and *z*, $C^{-1}=k_{B}T^{2}\Omega N$ and $k_{B}$ is Boltzmann’s constant, *h* the Planck’s constant, Ω the unit cell’s volume, and *N* the number of q-points. The summation in Equation (Equation 2) runs over all phonon modes λ; each mode has a frequency $\nu_{\lambda }$ and a group velocity $v_{\lambda }$, and at thermal equilibrium at temperature *T* its occupancy follows the Bose-Einstein distribution $f_{\lambda }$. The mean free displacement is initially calculated as $F_{j,\lambda }=\tau_{\lambda }v_{j,\lambda }$, where $\tau_{\lambda }$ is the lifetime of mode λ within the relaxation time approximation (RTA). Starting from this guess, the solution is then obtained iteratively and $F_{j,\lambda }$ takes the general form $\tau_{\lambda }\left(v_{j,\lambda }+\Delta_{j,\lambda }\right)$, where the correction $\Delta_{j,\lambda }$ captures the changes in the heat current deriving from the deviations in the phonon populations with respect to the solution at the RTA level and that result from the iterative process [32,33]. Scattering from isotopic disorder is also included considering the natural distributions of Ti and S isotopes within Tamura’s model [34].

## 3. Results and Discussion

Our first step is to evaluate the dependence of the thermal conductivity tensor on the number of **q**-points included in the solution of the BTE. This is a preliminary analysis necessary to assess the accuracy of our calculations. Figure 2 shows that, while the values of $\kappa_{xx}$ at 300 K are virtually insensitive to the number of **q**-points, we obtain variations of $\kappa_{yy}$ of less than 2.0% with a grid of least 10×10
**q**-points, if compared to the results obtained with a much thicker and computationally expensive 28×28 mesh. This accuracy is comparable with the dispersions typically reported by experimental measurements. We observe a similar behavior at 100 K, where convergence might be more difficult to achieve because of the larger relative weight of long-wavelength phonons. At this temperature we observe that $\kappa_{xx}$ oscillates at around an average value of 33.31 W m${}^{-1}$ K${}^{-1}$ for all grid resolutions, whereas $\kappa_{yy}$ requires at least a mesh of 16×16
**q**-points to stabilize with fluctuations smaller than 7% for finer meshes. Based on these convergence tests, hereinafter we present all our results in this manuscripts using a mesh of 16×16
**q**-points for the monolayer and bilayer and 16×16×16
**q**-points for the bulk phase, which both give an optimal compromise between accuracy and computational load.

We next illustrate in Figure 3 the dispersion relations calculated using the second-order force constants for these three structures. The most striking finding is that the lowest acoustic branches near the Γ point exhibit a *pocket* of negative frequencies along the Γ-Y direction for both nanosheets. For the monolayer, this pocket yields imaginary frequencies as low as −0.19 THz at **q** = (0.06, 0.0, 0.0) and for the bilayer, this singularity is extended farther away from Γ with lower frequencies that decrease down to −0.91 THz at **q** = (0.13, 0.0, 0.0). The increased number of low-frequency bands in close proximity to one another in the case of the bilayer makes the artifact worse by increasing the sensitivity of the results to spurious contamination between polarizations, but the problem completely disappears in the 3D solid once the monolayer bands are hybridized to give the familiar bulk acoustic branches. These imaginary frequencies appear because the flexural phonon band near Γ is inaccurately described, as the periodic boundary conditions imposed along all three axes in DFT calculations of layered systems cause a breakdown of the continuous rotation symmetry at the root of that quadratic character of the ZA branch. [18,35] Instead, this band should have a quadratic dispersion with a vanishing derivative in *q*, i.e., ∂ω/∂q→0.

The following step is then to enforce rotational symmetry by applying the correction proposed in Ref. [18]. This correction adds a post-processing step to the second-order force-constant calculation, where it proceeds by systematically building a basis of independent internal coordinates (distances, angles, and dihedrals) that are explicitly scalar, i.e., rotationally invariant. The Cartesian force constants are then projected on that basis and back. Since those internal coordinates can express any atomic movement except for the three continuous translations and the three continuous rotations, the result is compatible with the homogeneity and isotropy of free space by construction, leading to the expected quadratic behavior of the flexural branch. As shown in Figure 3a,b, the pocket is also exhibited by the other acoustic bands close to Γ, due to cross-contamination between polarizations; the correction solves the issues for those bands too, which is especially noticeable in the case of the bilayer. After correcting the second-order force constants, the pocket of negative frequencies disappears and the flexural band exhibits the expected quadratic dependence in the full dispersion relations illustrated in Figure 3c,d.

With the corrected harmonic force constants, we now calculate the thermal conductivities for both nanosheets as a function of temperature between 100 K and 600 K under the RTA and by solving the full BTE. Figure 4 indicates that this quantity decreases with decreasing temperature, indicating that Umklapp processes are the dominant scattering mechanism for the RTA and the full iterative solution. We also find that solving the full BTE produces higher conductivities than those determined under the RTA. For instance, this increase for the monolayer is 29.40% for $\kappa_{xx}$ and 13.34% for $\kappa_{yy}$ at 300 K. This enhancement gives an estimate of the role of momentum conserving Normal processes as a function of temperature and the underestimation of RTA derives from the fact that they are erroneously considered as resistive collisions. Furthermore, we evaluate the influence of symmetrizing the IFCs on the thermal conductivity. Figure 4b,d show that only the *x*-component is significantly affected by this correction since the pocket of imaginary frequencies appears in the Γ-Y direction with variations of 6.24 % for the monolayer and −6.44 % for the bilayer at 300 K.

Our results evidence a significant anisotropy of thermal transport in layered TiS${}_{3}$, as opposed to other 2D materials such as graphene [36] or transition metal dichalcogenides [35,37]. For instance, our BTE values at 300 K are $\kappa_{xx}=8.24$ W m${}^{-1}$ K${}^{-1}$ and $\kappa_{yy}=16.28$ W m${}^{-1}$ K${}^{-1}$ for the monolayer and $\kappa_{xx}=6.36$ W m${}^{-1}$ K${}^{-1}$ and $\kappa_{yy}=11.57$ W m${}^{-1}$ K${}^{-1}$ for the bilayer, yielding anisotropy ratios $\kappa_{yy}/\kappa_{xx}$ of 1.98 for the former and 1.82 for the latter. For the bulk, we find BTE values at 300 K are $\kappa_{xx}=8.28$ W m${}^{-1}$ K${}^{-1}$ and $\kappa_{yy}=14.34$ W m${}^{-1}$ K${}^{-1}$ (see Figure 5). We believe that this anisotropy in all three systems is a consequence of the atomic structure of TiS${}_{3}$ that, to some extent, can be considered as consisting of covalently bonded chains along the *y*-axis, with weaker interchain couplings along the *x*-axis. Furthermore, our calculations suggest that the thermal conductivity reduction when doubling the monolayer thickness is caused by the van der Waals forces. Although these forces are weak, they create small distortions in the atomic structure of the two joined layers that perturb the original degeneracy of their phonon branches. As a result, this perturbation increases the phase space for anharmonic scattering and hence its rates. This trend is inverted when moving from the bilayer to the bulk, where the thermal conductivity increases and the values of the monolayer are almost recovered. In this case the higher symmetry of an infinitely thick multilayer (i.e., the bulk), narrows again the phase space and thus the thermal conductivity can increase. As expected, we predict a much more inefficient thermal transport in the out-of-plane direction, where phonon propagation is mediated by the low van der Waals interlayer interaction, with values of $\kappa_{zz}$ that are 7 times smaller than those of $\kappa_{xx}$ and 12 times than those of $\kappa_{yy}$.

Last but not least, we evaluate the contributions of all phonon frequencies to the thermal conductivity. Figure 6a illustrates that the component $\kappa_{xx}$ plateaus for all three systems at around 5 THz. At this frequency, the individual contributions drop to a value below 0.1 W m${}^{-1}$ K${}^{-1}$ THz${}^{-1}$ given that most of the heat is transferred by the acoustic branches drawn in Figure 3a,b. In contrast, the component $\kappa_{yy}$ for both layers plateaus near 7 THz as the second-lowest lying acoustic branch is slightly steeper near Γ in the direction Γ-Z than along the Γ-Y. This variation produces a slightly higher group velocity, which in turn leads to higher contributions to the thermal conductivity. The plots of Figure 6 also reveal that the higher thermal conductivity of the monolayer essentially derives from the larger contribution of low-frequency phonons, particularly in the case of $\kappa_{xx}$ where its cumulative value dramatically rises at around 1 THz.

## 4. Conclusions

In conclusion, we have investigated the thermal transport in monolayers and bilayers of titanium trisulphide as well as in its bulk phase with first-principles calculations. The second-order force constants are corrected to enforce rotational symmetry, which has been recently shown to be lost by imposing periodic boundary conditions in systems with vacuum gaps. Our results indicate that the thermal conductivity is highly anisotropic with values at 300 K of 8.24 W m${}^{-1}$ K${}^{-1}$ in the *x* direction and 16.28 W m${}^{-1}$ K${}^{-1}$ along the *y*-axis for the monolayer. For the bilayer, our model predicts significant reductions of 22.72 % with 6.36 W m${}^{-1}$ K${}^{-1}$ in the *x* direction and 28.95 % with 11.57 W m${}^{-1}$ K${}^{-1}$ along the *y*-axis. This decrease is attributed to the van der Waals forces that introduce small coupling between the modes of both monolayers, slightly break the degeneracy of the phonon bands, and therefore increase the phase space for anharmonic scattering. In addition, these in-plane conductivities show the highly anisotropic nature of thermal transport in layered titanium trisulphide with ratios of 1.98 for monolayer and 1.82 for the bilayer. We believe that our results will help the community to recognize the importance of taking into account the loss of rotational symmetry in first-principle calculations of 2D systems with a vacuum gap, as well as to compare more carefully numerical data from monolayers and the experimental measurements taken samples with thicknesses of several layers. These results are also important for emerging nanodevices based on 2D materials and many layered van der Waals crystals, which are expected to play an important role in several applications, ranging from optoelectronics to thermoelectricity.

## Figures and Tables

**Figure 1 nanomaterials-10-00704-f001:**
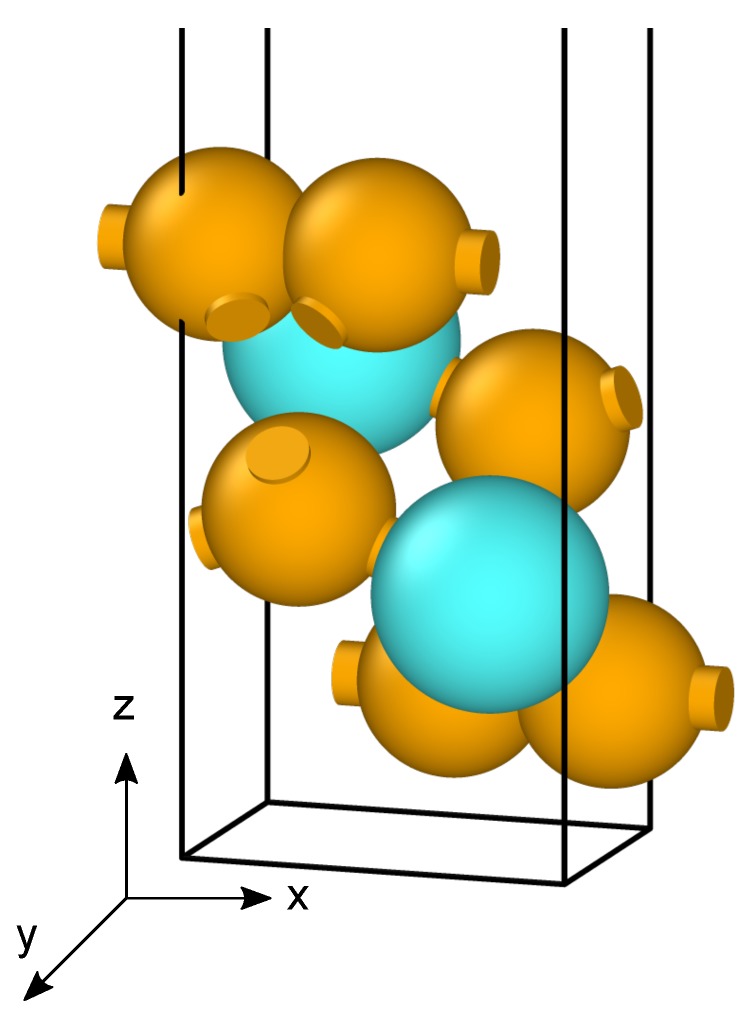
Representation of a TiS${}_{3}$ unit cell with atoms painted in blue for titanium and in orange for sulfur.

**Figure 2 nanomaterials-10-00704-f002:**
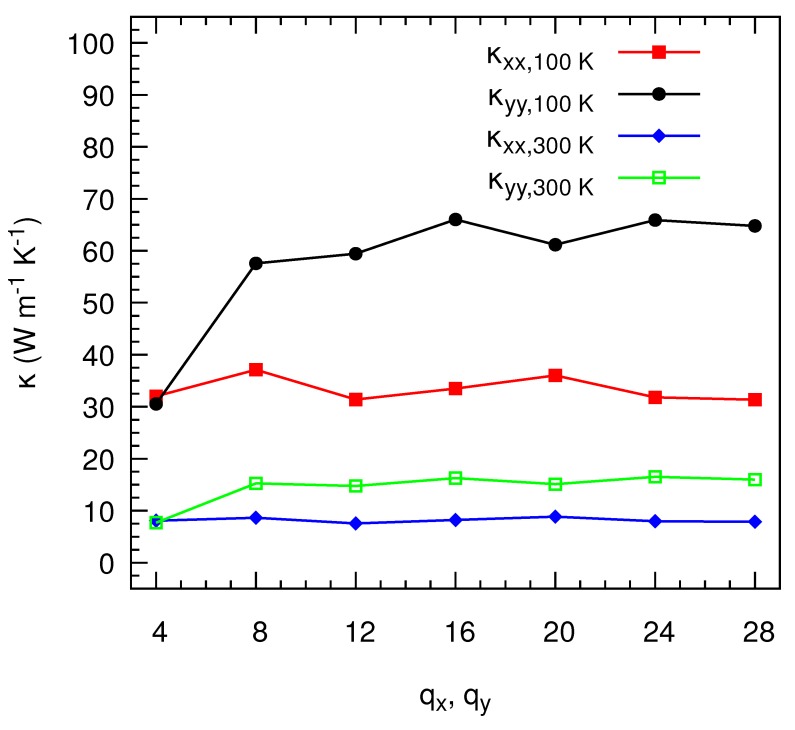
Thermal conductivity as a function of the number of **q**-points in the x (q${}_{x}$) and y (q${}_{y}$) directions at 100 K and 300 K.

**Figure 3 nanomaterials-10-00704-f003:**
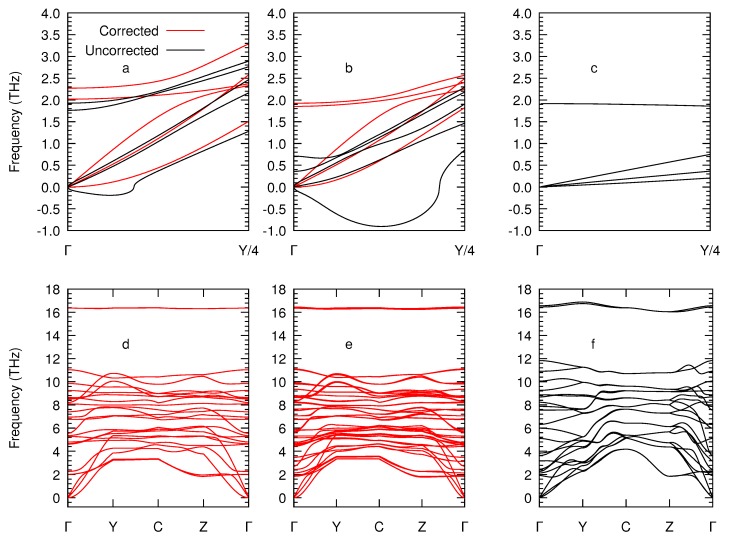
Zoomed phonon dispersions along the Γ-Y direction before and after correcting the harmonic IFCs according to the procedure given in Ref. [18] for the monolayer (**a**) and bilayer (**b**) as well as the uncorrected spectra for the bulk (**c**). Full corrected dispersion relations for the monolayer (**d**) and the bilayer (**e**) and uncorrected for the bulk (**f**).

**Figure 4 nanomaterials-10-00704-f004:**
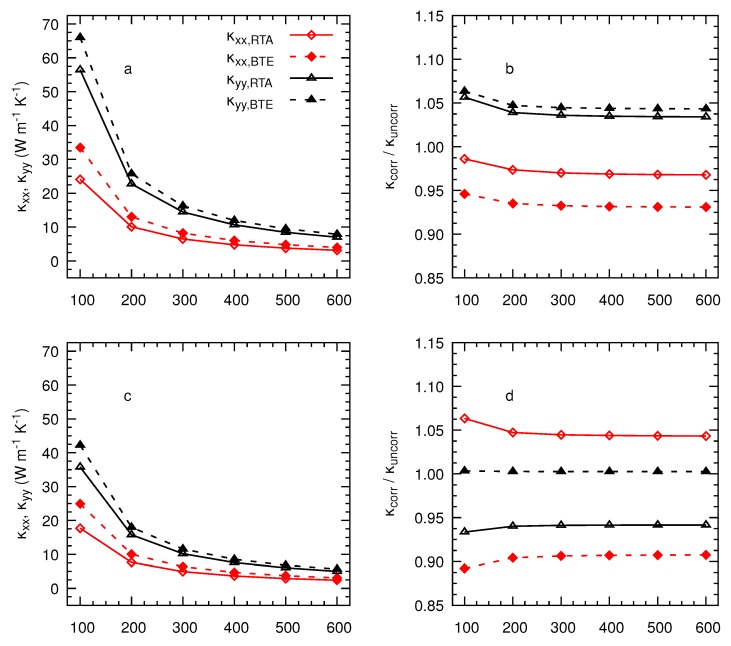
Corrected values of the thermal conductivity as a function of temperature after correcting the IFCs of the TiS${}_{3}$ monolayer (**a**) and bilayer (**b**). Ratios of the corrected thermal conductivity with respect to their uncorrected values for the monolayer (**c**) and bilayer (**d**).

**Figure 5 nanomaterials-10-00704-f005:**
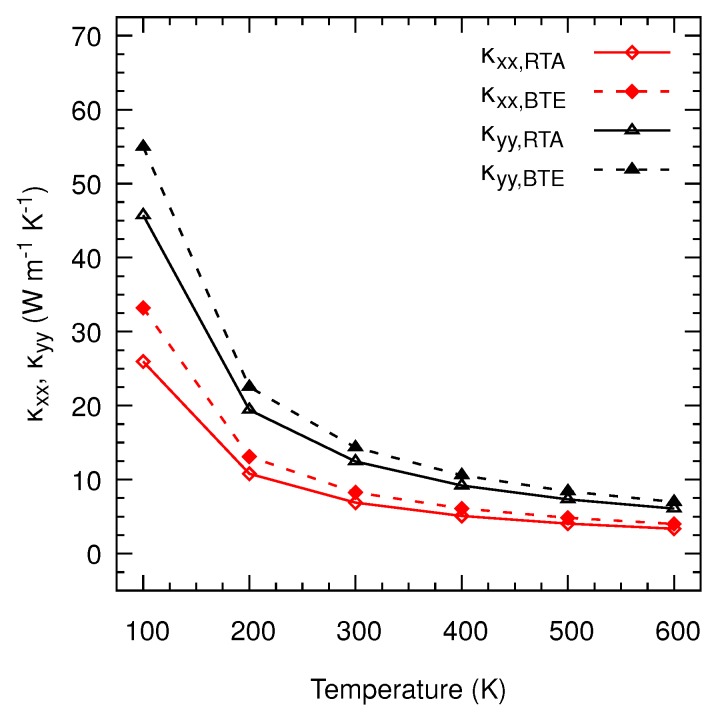
Evolution of the components of the thermal conductivity tensor as a function of temperature of bulk TiS${}_{3}$.

**Figure 6 nanomaterials-10-00704-f006:**
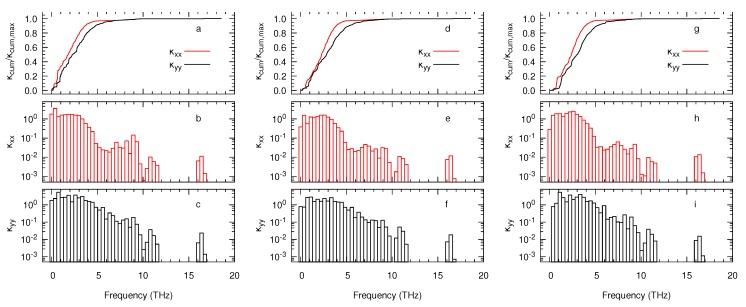
Modal decomposition of the components of the thermal conductivity tensor $\kappa_{xx}$ and $\kappa_{yy}$ (both in W m${}^{-1}$ K${}^{-1}$ THz${}^{-1}$) for the monolayer (**a**–**c**), bilayer (**d**–**f**), and bulk (**g**–**i**) with respect to the phonon frequency. Panels (**a**,**d**,**g**) show the normalised cumulative function of both components.

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
