# Peer review of "Anisotropic Thermal Conductivity in Few-Layer and Bulk Titanium Trisulphide from First Principles"

_nanomaterials, 2020, doi:10.3390/nano10040704_

Round 1
Reviewer 1 Report
Accept in present form!
Author Response
We thank the referee for the positive evaluation of our work.
Reviewer 2 Report
The authors calculate the thermal conductivity tensor of the 1L, 2L and bulk TiS3, to check if one can obtain an increase of the figure of merit of the materials due to possible anisotropy of the thermal conductivity.
They demonstrated that one has to implement the translational and rotational symmetries, in order to avoid unphysical solutions for the phonon spectrum. Probably, the most important result of the paper is stronger thermal transport in y direction in all studied systems. The authors propose that the reason for this is that the covalently-bound chains are stronger bound in y direction. The authors also found that the thermal conductivity decreases as one goes from 2L to 1L system, due to an increased anharmonicity scattering space in the two-layer case.
The paper is well-written, details of the calculations are provided, including the number of k-points needed top obtain converged results.
Thus, I recommend the paper for publication after the authors take into account the following minor points:
- p.3, line 79 to define how "deviations in the phonon populations" were calculated
- p.2, line 25 pf --> of
- p.3, line 78 add index j in the last Delta_lambda
- p.3, line 81 istopes --> isotopes
- p.6, line 143 covalently --> covalently
Author Response
The paper is well-written, details of the calculations are provided, including the number of k-points needed top obtain converged results.
We thank the referee for the positive evaluation of our work.
Thus, I recommend the paper for publication after the authors take into account the following minor points:
p.3, line 79 to define how "deviations in the phonon populations" were calculated
The deviations of the phonon population are computed with the iterative method extensively described in Li et al., Comp. Phys. Commun. 2014, 185, 1747 and here cited as Ref.[30].
To clarify this point we have rephrased the explanation given in text as follows:
“[...] Δj,λ captures the changes in the heat current deriving from the deviations in the phonon populations with respect to the solution at the RTA level and that result from the iterative process [30,31]”
p.2, line 25 pf --> of
p.3, line 78 add index j in the last Delta_lambda
p.3, line 81 istopes --> isotopes
p.6, line 143 covalently --> covalently
We have made all these corrections.
Reviewer 3 Report
This manuscript is dedicated to a study of the anisotropic thermal conductivity in few-layer and
bulk titanium trisulphide from first principles. The method and level of theory as used for the mentioned purpose are quite adequate.
Such research is quite timely, such material systems and their electronic and thermal properties are currently attracting quickly increasing research efforts.
There are very good figures and logically presented results from the simulations.
Manuscript is very complete and well written. I have just one suggestion related to the conclusions: by one or two lines added, the present results should be put in the broader context of the emerging nano-devices based on 2D and ultrathin materials. Such addition, although partly speculative, would strengthen further the final message of this excellent manuscript after publication.
Author Response
Manuscript is very complete and well written.
We thank the referee for the positive evaluation of our work.
I have just one suggestion related to the conclusions: by one or two lines added, the present results should be put in the broader context of the emerging nano-devices based on 2D and ultrathin materials. Such addition, although partly speculative, would strengthen further the final message of this excellent manuscript after publication.
We have added the following sentence:
“These results are also important for emerging nanodevices based on 2D materials and many layered van der Waals crystals, which are expected to play an important role in several applications, ranging from optoelectronics to thermoelectricity.”